# MircoRNA in Extracellular Vesicles from Patients with Pulmonary Arterial Hypertension Alters Endothelial Angiogenic Response

**DOI:** 10.3390/ijms231911964

**Published:** 2022-10-08

**Authors:** Avinash Khandagale, Padraic Corcoran, Maryam Nikpour, Anders Isaksson, Gerhard Wikström, Agneta Siegbahn, Christina Christersson

**Affiliations:** 1Cardiology Section, Department of Medical Sciences, Uppsala University, SE 75185 Uppsala, Sweden; 2Clinical Chemistry Section, Department of Medical Sciences, Uppsala University, SE 75185 Uppsala, Sweden; 3Array & Analysis Facility Section, Department of Medical Sciences, Uppsala University, SE 75185 Uppsala, Sweden; 4Internal Medicine Section, Department of Medical Sciences, Uppsala University, SE 75185 Uppsala, Sweden

**Keywords:** extracellular vesicles, microRNA, pulmonary arterial hypertension, pulmonary artery endothelial cells, miR-486-5p, miR-26a-5p, angiogenesis, proliferation

## Abstract

Pulmonary arterial hypertension (PAH) is characterized by a progressive elevation of pulmonary pressure leading to right ventricular dysfunction and is associated with a poor prognosis. Patients with PAH have increased numbers of circulating extracellular vesicles (EVs) and altered expression of circulating microRNAs (miRs). The study aimed to evaluate the miR profile contained within purified EVs derived from the plasma of PAH patients as compared to healthy controls (HC). Circulating EVs, purified from platelet-free plasma were analyzed using flow cytometry, western blot, and electron microscopy. Total RNA isolated from EVs was subjected to Microarray analysis using GeneChip miRNA 4.0 Array and bioinformatics tools. Overexpression and inhibition of miRs were conducted in human pulmonary artery endothelial cells (hPAECs) that had been incubated previously with either PAH- or HC-derived EVs. Cell proliferation (MTT assay) and angiogenesis (tube formation assay) were tested in hPAECs to determine miR functionality. MiR profiling revealed 370 heats while comparing PAH and HC groups, 22 of which were found to be down-regulated and 6 were up-regulated in the PAH EVs. Among the altered miRs, miR-486-5p was overexpressed, while miR-26a-5p was downregulated in PAH EVs compared to HC EVs. Inhibition of mir-486-5p or overexpression of miR-26a-5p in hPAECs post-exposure of PAH EVs abrogated proangiogenic and proliferative effects posed by PAH EVs contrary to HC EVs. The angiogenic and proliferative effects of the miRs from PAH EVs were observed to be mediated through nuclear factor (NF)-κB activation. PAH EVs carry and present an altered miR profile that can be targeted to restrict angiogenesis and reduce pulmonary endothelium activation. Further studies concerning miRs from circulating heterogeneous EVs in PAH patients are warranted to understand their potential as targets for treatment in PAH.

## 1. Introduction

Pulmonary arterial hypertension (PAH) is a rare but progressive cardiopulmonary disease characterized by excessive vascular endothelial and smooth muscle cell proliferation, inflammation, and fibrosis, which leads to right heart failure and premature death [1]. The complex molecular mechanisms of PAH are not fully understood and include the influence of both common [2] and rare genetic variation, as well as micro RNAs (miRs) dysregulation [3].

At the molecular level, various cell types in the pulmonary arteries participate in remodeling events in PAH that influence the pathological manifestation in the pulmonary vessel wall. A hypoxic state, inflammation, vessel injury, and oxidative stress contribute to the remodeling process [4]. Despite advances in modern pulmonary hypertension-specific therapy, the mortality of PAH patients remains high, ranging between 22.2% and 54.5% [5].

One of the most sought-after mechanisms that have been increasingly investigated in recent times for the development of PAH is the role of extracellular vesicles (EVs). Circulating EVs are increased in PAH patients [6], and the amount of EVs correlates with pulmonary vascular resistance [7], functional impairment [8], and mortality [9]. EVs are essential mediators of cell-to-cell communication-carrying biomolecules such as proteins, DNA, messenger RNA, and miRs [10]. EVs can act as positive or negative modulators of cardiovascular diseases depending on the type and state of the cells from which they originate [11].

MiRs are small, non-coding RNA molecules found in tissues, plasma, and EVs. Among extracellular vesicles, miRs are differentially sorted into exosomes based on the presence of specific motifs in their sequence [12]. They are dysregulated in PAH and contribute to the disease process in animal models [13,14,15]. Whether miRs identified in PAH patients are positive or negative modulators of PAH is currently a subject of research. However, with increasing numbers of miRs found to be regulating biological processes [16], identifying their molecular function in disease progression is challenging.

In this study, we identify miRs associated with plasma-derived EVs from PAH patients and decipher their functional impact on cells involved in PAH pathogenicity. We assess the miRs expression within EVs from PAH plasma in comparison to healthy individuals and address the functional effects of selected miRs, which are highly altered in PAH, on human pulmonary artery endothelial cells.

## 2. Results

### 2.1. Characterization of Purified Extracellular Vesicles

In order to profile vesicles associated miRs, we purified EVs from PAH patients and healthy control individuals (HC) plasma. These EVs were then subjected to RNA purification and subsequent miR profiling using GeneChip^TM^ miRNA array analysis (Figure 1A). We demonstrated the presence of EVs, including exosomes in plasma derived from PAH and HC, by morphological and molecular characterization. Molecular characterization was performed by flow cytometry and western blot analysis, whereas morphological characterization was conducted in terms of Transmission Electron Microscopy (TEM) analysis. Both flow cytometry (Figure 1B) and western blot (Figure 1C) analysis for CD63 revealed the presence of exosomes within the purified EVs samples among both PAH and HC groups. TEM imaging confirmed the presence of vesicles of spherical shape with diameters ranging from 50 to 1000 nm, indicating a heterogeneous population of EVs compatible with its size from both PAH and HC groups (Figure 1D).

### 2.2. MicroRNAs Identification in EVs from PAH Patients and HC

Micro-array derived expression profile revealed a total of 370 miRs in the purified EVs from PAH patients and HC individuals. Of these, 28 (7.57% of the total) were significantly altered in PAH EVs, when compared to HC EVs (Figure 2A). HC and PAH samples were analyzed for microarray on more than one occasion; therefore, the expected batch effect was countered to obtain batch-independent sample distribution (Figure 2B). While 22 miRs were found downregulated, miR-6732-5p, miR-486-5p, miR-4793-5p, miR-6722, miR-320e, and miR-8075 were upregulated within the PAH group (Table 1).

Hierarchical clustering based on significant fold change of miR expression among PAH group divulged miscellaneous miR population (Figure 2C). Intriguingly, none of these miRs have been previously associated with PAH pathogenicity [17]. Gene ontology (GO) for biological processes analysis of these miRs did not result in any significant over-represented GO terms. However, it did show positive regulation of angiogenesis and sprouting angiogenesis high up on the list of results sorted by *p*-value (Appendix A). Interestingly, our previous publication also identified the proangiogenic potential of PAH EVs [6].

### 2.3. PAH EVs Derived miRs Effect on hPAECs

Next, we asked whether previously reported proangiogenic effects of PAH EVs were miRs derived. To address this question, we first explored whether EVs were spontaneously internalized by human pulmonary artery endothelial cells (hPAECs) and whether EVs-derived miRs can be altered within hPAECs. To address this, EVs were labeled with lipid-based membrane dye PKH67, followed by their incubation with hPAECs. Post-incubation cells were labeled with membrane dye FM4-64, followed by fixation. Confocal microscopy showed that labeled EVs, both PAH- and HC-derived, were internalized and appeared along the plasma membrane, as well as within the cytoplasm in the same 0.5-μm section as the lipophilic FM4-64 dye (Figure 3A). After a literature search, we selected miR-486-5p and miR-26a-5p, both were associated with angiogenesis in different pathologic conditions [18,19] but not PAH, for their functional role in hPAECs. As miR-486-5p was up-regulated and miR-26a-5p down-regulated in PAH EVs, we performed miR-inhibitor and -mimic transfection, respectively within hPAECs that were incubated with either PAH- or HC- EVs. In order to test miR transfection efficiency, *twf-1* gene expression was checked within miR-1 (positive control for transfection) transfected cells. As miR-1 is a negative regulator of the *twf-1* gene, downregulation of TWF-1 transcripts was determined where transfection efficiency was observed to be more than 65% (Appendix A). Both miR-486-5p inhibitor and miR-26a-5p mimic transfection resulted in significant miR-486-5p down-regulation and miR-26a-5p up-regulation respectively within hPAECs (Figure 3B,C). While miR-486-5p inhibition was observed only in hPAECs that were pre-incubated with PAH EVs (Figure 3B), miR-26a-5p upregulation was observed irrespective of the presence of either type of EVs (Figure 3C). Next, to test whether alteration of PAH EVs derived miRs within hPAECs influenced PAH EVs proangiogenic effect, we checked transcription and translation of VEGF-A, the prototypical member of the VEGF family, as a proangiogenic marker. VEGF-A mRNA showed significantly higher levels in PAH EV-stimulated hPAECs post 48 h of mock transfection as compared to HC EV-stimulated hPAECs (Figure 3D). This induction in VEGF transcription was abrogated when PAH EV-stimulated hPAECs were transfected with either miR-486-5p inhibitor or miR-26a-5p mimic (Figure 3D). Similarly, secretion of VEGF by hPAECs upon PAH EVs incubation was also significantly affected by miR-486-5p inhibitor or miR-26a-5p mimic transfection (Figure 3E). Increased cellular expression of VEGF by PAH EVs, was also abrogated by miR-486-5p inhibitor or miR-26a-5p mimic transfection (Figure 3F) Taken together, these data suggest PAH EVs-derived miRs positively affect proangiogenic signaling in hPAECs.

### 2.4. Effect of PAH EVs Derived miRs on hPAEC Tube Formation In Vitro

The effect of purified PAH EVs on hPAEC tube formation was examined using an in vitro Matrigel assay. PAH EVs with their miR cargo, induced a complex network of branched angiotubes, unlike HC EVs (Figure 4A). This effect remained unaffected if cells were mock-transfected for 72 h post EVs pre-incubation. However, when miR-486-5p inhibitor or miR-26a-5p mimic was introduced to hPAECs, this angiotube formation efficiency by PAH EVs was severely hampered (Figure 4A). Neither mock-transfected nor miR reagents-transfected hPAECs that were pre-incubated with HC EVs displayed any tube-like formation on the Matrigel assay (Figure 4A). Parameters of angiotubes, such as total tube length, number of loops, and the average loop area, were significantly higher in the hPAECs incubated with PAH EVs compared to their HC counterparts that were abrogated upon introduction of miR-486-5p inhibitor or miR-26a-5p mimic (Figure 4B–D). Similarly, cell proliferation was analyzed in hPAECs that were pre-incubated with either PAH EVs or HC EVs, followed by transfection by miR inhibitor or mimic for an additional 48 h. Cell activation, in terms of proliferation, was also shown to be driven by miRs from PAH EVs, as the presence of miR-486-5p inhibitor or miR-26a-5p mimic inhibits PAH EVs mediated hPAECs proliferation (Figure 4E). These results suggest that proangiogenic and proliferative effects rendered by PAH EVs over pulmonary endothelium were partly mediated through their miRs.

### 2.5. PAH EVs-Derived miRs Influenced NF-kB-Mediated Cellular Signaling

Next, we explored the targets of PAH EVs-derived miR-486-5p and miR-26-5p influencing proangiogenic and proliferative effects on hPAECs. PTEN and protein kinases have previously been shown to be involved either directly or through signaling by the selected miRs. In our studies both PTEN and MAP3K in hPAECs remained unaffected by pre-incubation of PAH EVs in the presence or absence of miR-486-5p inhibitor or miR-26a-5p mimic (Appendix A). Pathway analysis involving expression and gene reporter for nuclear factor kappa B (NF-kB) indicated PAH EVs-derived miRs impacted NF-kB signaling through miR-486-5p and miR-26a-5p (Figure 5). PAH EVs induced cellular NF-kB-p65 expression, and this increase was restricted when PAH EVs-incubated hPAECs were transfected with miR-486-5p inhibitor or miR-26-5p mimic (Figure 5A). Densitometry analysis of NF-kB band intensity revealed significant inhibition of PAH EVs-derived NF-kB expression by miR-486-5p inhibitor or miR-26-5p mimic (Figure 5B). Next, to check whether PAH EVs derived proangiogenic response by hPAECs was mediated by NF-kB, we inhibited NF-kB activity using irreversible inhibitor Bay 11-7085. It was observed that NF-kB inhibition indeed reversed PAH EVs contemplated VEGF release by hPAECs (Figure 5C). Similarly, to confirm the impact of miR-486-5p inhibitor and miR-26a-5p mimic on NF-κB signaling, hPAECs were first incubated with either PAH- or HC- EVs followed by transfection of miR inhibitor or mimic along with co-transfection of luciferase reporter or its negative control that contained multiple NF-κB binding sites in its promoter. Compared with the HC EVs, PAH EVs-incubated hPAECs displayed a marked increase in NF-κB promoter activity measured in terms of luciferase intensity, which was significantly restricted in response to miR-486-5p-inhibitor and miR-26a-5p mimic transfection to PAH EVs exposed hPAECs (Figure 5D). Taken together, these results indicate that the NF-kB signaling route taken by PAH EVs-derived cargo to promote angiogenesis in hPAECs was partly mediated positively by miR-486-5p and negatively by miR-26a-5p.

## 3. Discussion

In the present study, we identified novel microRNAs (miRs) carried by endogenous circulating extracellular vesicles (EVs) from the plasma of PAH patients and stipulated their biological role in pulmonary endothelium. Of the total 370 identified miRs, 28 were found differentially expressed in PAH EVs. We not only compared the expression profiles of miRs from circulating EVs in PAH patients and healthy controls but also shed light on some of the altered miRs putative roles in the progression of PAH pathophysiology.

EVs, both microvesicles and exosomes harbor miRs. As a heterogeneous group of vesicles, they exert their effects in the circulation through uptake into target cells in the vessel wall by depositing their vesicular contents [20]. We demonstrated that plasma derived heterogeneous EVs from PAH patients induce endothelial dysfunction and promote endothelial angiogenesis and proliferation [6] while others have shown that PAH EVs release their contents to modulate mitochondrial functions [21].

It has been widely demonstrated that miRs are detectable in plasma and that circulating miRs may have the potential to serve as new noninvasive biomarkers in patients with PAH [22]. Increasing evidence suggests that alteration in miRs expression level can lead to various cardiovascular disorders, including PAH [23,24]. Additionally, some studies reported over 150 plasma-derived miRs [25] while others reported only a few [26,27,28] being dysregulated in PAH. Increasing evidence indicated more research analyzing the expression profiles of miRs and their signaling pathway in cells of the pulmonary vasculature, animal models of PAH, and PAH patients [17]. In rodent models of hypoxia-induced PAH, suppression of miR-204, miR-22, miR-30, and let-7f while up-regulation of miR-322 and miR-451 was observed [26,29]. Recently, miR-146-5p/USP3 axis was postulated as a possible target for PAH treatment based on the evidence that miR-146-5p promoted hPAECs proliferation under hypoxic conditions through targeting USP3 [30]. Also, the miR143/145 cluster regulates apoptosis and the proliferation of arterial smooth muscle cells during experimental PAH [31]. Despite all this data, only a handful of circulating miRs have been confirmed to play a role in PAH using a variety of experimental models of PAH [22], whereas there is limited knowledge regarding EVs-derived miRs in this regard. These findings prompted us to investigate the usefulness of circulating miRs contained within EVs in patients with PAH.

In the present study, we can confirm some of the previous reports and have identified altered expression of miRs in EVs derived from the plasma of PAH patients and healthy individuals. We investigated PAH EV-derived miRs expression and found significant changes in 28 miRs, as supported by micro-array analysis, followed by hierarchical clustering and PCA analysis, most of which (22) were down-regulated, whereas the remaining 6 were up-regulated in PAH. Although no GO term was significantly over-represented by differentially expressed miRs using GO-biological processes analysis, nonetheless, angiogenesis, relevant during PAH development, appeared high up on the list of results sorted by *p*-value. Moreover, none of the differentially expressed miRs have thus far been found to be directly associated with PAH pathology. Nevertheless, some of the miRs that are found to be affected in PAH EVs are known to be involved in other lung diseases; for example, miR-342, found to be down-regulated in PAH EVs in the present study, has been shown to exacerbate neonatal broncho-pulmonary dysplasia [32]. Likewise, miR-486 carrying EVs was more recently exhibited to promote angiogenesis after myocardial infarction in mice [18], whereas miR-26a, found to be upregulated in the present study, was identified as inhibiting angiogenesis in a cellular model of hepatic carcinoma [33].

It is well known that proangiogenic response plays an important role in the development of PAH, and the inappropriate expression of angiogenic markers and their receptors is often associated with PAH progression [34]. Recent findings from our group also suggested a strong relation of PAH EVs in promoting endothelial angiogenesis [6]. From the list of 28 altered miRs from our microarray analysis over PAH EVs, we searched the literature and found only miR-486-5p, mir-26a and miR-342 associated with angiogenesis [35,36,37] and that none was in the context of PAH. When we tested these miRs in our cellular models, over-expression of miR-486-5p and under-expression of miR-26a-5p was found to be influential in terms of human pulmonary endothelial cell’s response to PAH EVs.

Studies have shown that PTEN, PIK3R1, and MMPs are potential downstream target genes of miR-26a and miR-486-5p [18,38,39,40]. However, in our studies, including PAH EVs-derived miRs, these genes PTEN and protein kinase (MAP73K7) remained unaltered. Instead, we identified NF-kB as a novel mediator of miR-486-5p and miR-26a-5p signaling. The nature and extent to which these miRs affect NF-kB signaling remain elusive, and are beyond the scope of the current study. Here, we first confirmed that miR-486-5p and miR-26a-5p expression is altered in circulating EVs from the plasma of PAH patients. Second, we determined whether these miRs influence PAH EVs-derived angiogenic and proliferative response within pulmonary endothelium and found that the inhibition of miR-486-5p and overexpression of miR-26a-5p led to the suppression of proliferation as well as VEGF transcripts and protein expression in PAH EVs-induced hPAECs. Moreover, there was a direct correlation of miR-486-5p and an inverse correlation of miR-26a-5p expression by PAH EVs with respect to proliferative and proangiogenic effects within hPAECs. Our results also suggest that these effects induced by miR-486-5p and miR-26a-5p are partly achieved by targeting NF-kB signaling.

Overall, the current study provided novel miRs that are found altered in purified EVs from PAH patient plasma and ex vivo functional evidence to extend their role in PAH pathogenicity. These results may help us understand and delineate the complex molecular mechanisms of PAH development and pave ways to further research involving miRs presented in EVs from PAH plasma to explore their therapeutic targeting in PAH pathobiology.

## 4. Methods

### 4.1. Patient Cohort and Sample Collection

A total of 70 patients with PAH were admitted to the outpatient clinic for PAH at Uppsala University Hospital, Sweden, and descriptions of these patients have previously been published [6]. Ten out of the 70 were included in the present study. Healthy individuals, similar numbers of females and males, without any medical treatment were asked to participate as healthy controls (HC, *n* = 10) and included during the same time period as the PAH patients (*n* = 10). The study was approved by the local ethics committee (ref. no. 2010/343) and was conducted in accordance with the ethical principles of the Declaration of Helsinki.

### 4.2. Blood Collection, Plasma Storage and Preparation of Extracellular Vesicles (EVs)

Whole blood from PAH patients and the HC group was collected in citrate tubes (3.8%) by direct puncture without stasis. The preparation of whole blood started within 30 min of collection. Samples were centrifuged twice at 2500× *g* for 15 min to get platelet free plasma which was further ultracentrifuged at 20,000× *g* for 60 min at room temperature. The supernatant was subsequently removed. The pellet was used as purified EVs and dissolved in 200 µL HEPES-buffered saline (HBS). The concentration of EVs were determined by the Lowry method (BioRad).

### 4.3. Flow Cytometry of Purified Extracellular Vesicles

Purified EVs (200–1000 nm) were characterized using a CytoFLEX flow cytometer (Beckman Coulter, Bromma, Sweden). Cytometer was standardized, calibrated using fluorescent silica beads (Megamix FSC & SSC Plus, BioCytex, Marseille, France), and gating parameters were set to measure extracellular vesicles as described previously [6]. Purified EVs from PAH and HC were incubated with APC-conjugated exosomes specific CD63 (clone H5C6; NordicBiosite, Täby, Sweden) antibody prepared in HBS and run through the cytometer for exosome detection. The flow cytometer was washed before and between samples using filtered (0.04 µm filter units) double-distilled water to avoid background noise. The remaining configurations and the settings were maintained according to the manufacturer’s recommendations for detection of EVs.

### 4.4. Transmission Electron Microscopy (TEM) of Negatively Stained EVs

The purified concentrated frozen total EVs were detected using TEM imaging technique, described elsewhere [20]. Briefly, frozen EVs were thawed and fixed before placing on a formvar- and carbon-coated grid. Samples were then washed and subsequently stained in Uranylacetate (UA) on ice for 10 min. Dried grids were examined by TEM (FEI Tecnaii G2; AMOLF, Amsterdam, The Netherlands) operated at 80 kV following the manufacturer’s instructions.

### 4.5. RNA Extraction from Extracellular Vesicles

Plasma EV RNA was isolated from 100 μL EVs sample (as prepared above: equivalent to EVs purified from 500 μL plasma) using Ambion mirVana™ miRNA Isolation Kit (ThrmoFisher Scientific, Göteborg, Sweden)or Invitrogen Total Exosome RNA & Protein Isolation Kit (ThrmoFisher Scientific, Göteborg, Sweden), following the supplier’s instructions. In each case, the EV pellet was initially re-suspended in kit-specific lysis solution with the addition of 100:1 β-mercaptoethanol where appropriate. RNA concentration was assessed using the NanoDrop (Thermo Scientific, Waltham, MA, USA).

### 4.6. MicroArray Analysis of microRNA from Purified Extracellular Vesicles

Fixed volumes (45 μL) rather than fixed amounts of total RNA from each sample were used to prepare biotinylated fragmented cRNA according to the FlashTag Biotin HSR RNA labeling kit (PN 703095 Rev. 1). 130 μL of each sample were loaded to the GeneChip miRNA 4.0 Array (Covering 2578 human mature miRs; ThermoFisher Scientific, Göteborg, Sweden) and then hybridized for 16–18 h in a 48 °C incubator, rotated at 60 rpm. According to the GeneChip TM Expression Wash, Stain and Scan Manual (PN 702731 Rev3, Affymetrix Inc., Santa Clara, CA, USA) the arrays were then washed and stained using the Fluidics Station 450 and finally scanned using the GeneChip TM Scanner 3000 7G.

### 4.7. Microarray Data Analysis

The raw data was normalized in the Transcriptome Analysis Console, version 4.0.2.15, provided by ThermoFisher (https://www.thermofisher.com/se/en/home/life-science/microarray-analysis/microarray-analysis-instruments-software-services/microarray-analysis-software/affymetrix-transcriptome-analysis-console-software.html (accessed on 13 January 2021)), using the Robust Multi-Array Average (RMA) method [41,42]. Annotations were obtained using the miRNA-4_0-st-v1.annotations.20160922.csv file. Subsequent analysis of the normalized expression data was carried out in the statistical computing language R version 4.0.3 (http://www.r-project.org (accessed on 13 January 2021)). The miRBaseConverter bioconductor package (v1.14.0) was used for converting miRNA names from v20 to the latest v22 miRBase ids [43]. Probe sets were filtered to exclude non-expressed miRNAs. A miRNA on the array was considered expressed if at least 9 samples (the size of the smallest group) had a detection above background (DABG) *p*-value < 0.05. Principal component analysis (PCA) and hierarchical clustering were performed and a batch factor was included in the design matrix used in the limma analysis to address the batch effect observed in the PCA. The prcomp function in R was used for the PCA and pheatmap (v1.0.12) was used for heatmap generation. In order to search for the differentially expressed miRNAs between the groups, an empirical Bayes moderated *t*-test was applied using the ‘limma’ package (v3.46.0) [44,45]. To address the problem of multiple testing, the *p*-values were adjusted using the method of Benjamini and Hochberg [46].

### 4.8. Functional Enrichment of the miRNA Targets and miRNA

Unbiased functional enrichment analysis of the miRNA gene targets was performed with BUFET [47], which performs an empirical enrichment analysis as outlined in Bleazard et al. (2015) [48]. The set of predicted miRNA gene interactions for humans was downloaded from mirDB v6.0 (http://mirdb.org/download.html (accessed on 19 January 2021)). Only predicted gene targets with a score greater than 80 were kept for functional enrichment analysis (http://mirdb.org/faq.html (accessed on 13 January 2021)). GO terms for the genes were downloaded from Ensembl v102 using biomart (accessed on 13 January 2021). The Kegg pathways to which genes are assigned were downloaded using from the KEGG database (https://www.kegg.jp/kegg/ (accessed on 13 January 2021)) using the KEGGREST R package. The curated gene-disease associations were downloaded from DisGeNeT (https://www.disgenet.org/downloads (accessed on 13 January 2021)). The BUFET analysis was run with 1 million iterations. The results from the empirical enrichment analysis were filtered to retain terms with an adjusted *p*-value less than <0.05 and to exclude GO terms, Kegg pathways, or disease gene sets with fewer than 5 genes.

A total of 1419 reference miRNA sets were downloaded from TAM 2.0 (http://www.lirmed.com/tam2/Public/static/data/mirset_v10.txt; accessed on 18 January 2021) and were composed of six categories, including function, disease (HMDD), family, cluster, tissue-specific, and transcription factor. The functional enrichment analysis of these categories was performed using the gost function from the gprofiler2 R package [49] taking the differentially expressed miRNAs as the query, and multiple test correction was performed using the gSCS method from gprofiler [50].

### 4.9. Cell Culture and miR Silencing or Overexpression

The human pulmonary artery endothelial cells (hPAECs) were maintained in phenol red-free endothelial cell (EC) growth media with supplements (PromoCell) in 5% CO2 at 37 °C with the highest density never exceeding 10^6^ cells/mL. To silence or overexpress EVs-derived miRs, hPAECs were seeded at a density of 200,000 cells/mL in 24 well plates (ThermoFisher Scientific, Göteborg, Sweden) for 24 h. Cells were then first incubated with EVs (50 μg/mL) from PAH or HC for 1 h in order to facilitate EVs internalization, followed by transfection of miR-486-5p inhibitor (100 nM; ThermoFisher Scientific) or miR26a-5p mimic (100 nM; ThermoFisher Scientific) or mock-transfection without miR reagent using miRVANA transfecting agent (ThermoFisher Scientific) in OptiMEM media (ThermoFisher Scientific) for additional 24 h. Post transfection OptiMEM was replaced with EC growth media with supplements, and cells were allowed to grow for a further 24 to 48 h before functional analysis.

### 4.10. EV Labeling and Intracellular Visualization Using Immunofluorescence

Purified EVs (50 μg/mL) from HC and PAH patients were labeled with PKH67 (Sigma-Aldrich, St. Louis, MO, USA) for 15 min at room temperature (modified from [51]). Before labeling, the antibody and dye solution was centrifuged at high speed for 15 min at 4 °C and the supernatant was further filtered through 0.2 μ filter units to remove any aggregates. A mixture without EVs was used as a control for detecting any carryover of labeled antibodies within the cells. HPAECs (PromoCell, Heidelberg, Germany) were seeded at a density of 1 × 10^5^ cells/mL in ibidi μ-slide (ibidi GmbH, Germany) for 24 h at 37 °C. The next day the cell monolayer was incubated with labeled EV solution for 1 h at 37 °C. Post-45 min with EVs incubation the cells were also labeled with FM^TM^4-64FX cell-membrane labeling solution (Thermo Scientific) prepared in phenol red-free cell growth media for another 15 min at room temperature in the dark. Subsequently, cells were washed with PBS and fixed in 2% paraformaldehyde solution for 30 min. After fixation, μ-slides were washed and mounted using ProLong Gold Antifade Mountant containing DAPI (Thermo Fisher Scientific). Images were obtained using a ZEISS LSM 710 confocal microscope (Carl Zeiss, Oberkochen, Germany).

### 4.11. RT-PCR Analysis of EV-Derived miRs and Angiogenesis Biomarkers within hPAECs

Expression profiling of selected miRs identified through MicroArray analysis was performed using an Applied Biosystems 7900HT real-time PCR instrument. Briefly, RNA samples from hPAECs that were incubated with PAH- or HC-derived EVs and mock-transfected or transfected with miR inhibitor or miR mimic. The RNA was transcribed using miRNA-specific stem-loop primers (TaqMan MicroRNA Assays; Applied Biosystems), Post reverse transcription, real-time quantitative PCR (qRT-PCR) was performed using miR-expression assays for miR-486-5p and miR-26a-5p (TaqMan MicroRNA Assays; Applied Biosystems, ThermoFisher Scientific, Göteborg, Sweden) and the cycle number at which the reaction crossed a threshold (CT) was determined for each gene. The expression level of miRs was evaluated by a comparative CT method using GAPDH median normalization. There are no genes that are known to be expressed with the same copy number in EVs that could be used as normalization controls. Thus, raw CT values were normalized using a median CT value (ΔCT = CTmiRNA − CTmedian), and the relative amount of miR in EVs relative to hPAEC’s GAPDH (fold change) was described.

For VEGF gene expression in hPAECs, cells were incubated with or without EVs purified from HC or PAH patients for 1 h at 37 °C, followed by transfection of miR-486-5p inhibitor or miR-26a-5p mimic for 48–72 h. Cells were then washed and lysed in a lysis buffer provided in the kit, and mRNA was purified using the QIAGEN RNeasy Mini Kit according to the manufacturer’s protocol. Post reverse transcription, the expression of VEGF and GAPDH were measured by qRT-PCR using assay-on-demand primers and probes (ThermoFisher Scientific, Göteborg, Sweden) on an AbiPrism 7500 system and analyzed using the 7500 Fast software (Applied Biosystems, ThermoFisher Scientific, Göteborg, Sweden). Data are shown as relative mRNA transcripts normalized to the expression of the GAPDH housekeeping gene.

### 4.12. VEGF ELISA from Cell Culture Media

Cell culture supernatant samples from different cell passages of hPAECs with or without EVs post 72 h of miR inhibitor or mimic transfection and with or without Bay 11-7085 (NF-kB inhibitor; 100 nM for 24 h; Sigma-Aldrich, Stockholm, Sweden AB) were analyzed by electrochemiluminescence using the VEGF Human ELISA kit (ThermoFisher Scientific, Göteborg, Sweden) following manufacturer’s instructions. Data is represented as pg/mL of VEGF based on the standard provided in the kit.

### 4.13. Tube Formation Assay

Angiogenesis tube formation assay was performed as mentioned elsewhere [6]. Briefly, the required number of hPAECs were seeded in a 24-well plate overnight at 37 °C in a 5% CO_2_-humidified incubator. The next day cells were stimulated with or without purified EVs from HC or PAH patients for 1 h followed by mock-transfection or transfection of miR-486-5p inhibitor or miR-26a-5p mimic for 48–72 h. Simultaneously, Matrigel basement membrane matrix (growth factor reduced; BD Biosciences, Stockholm, Sweden)) polymerized in ibidi angiogenesis slides (Ibidi, Gräfelfing, Germany). Treated cells were trypsinized and seeded onto polymerized Matrigel and observed for 4 h and 8 h for tube-formation capacity. Tube formation was observed and photographed using a microscope-compatible camera (Nikon). Images were further analyzed and interpreted using ACAS image analysis software (MetaVi Labs, Ibidi, Germany) based on different parameters such as branching points, tube lengths, tube numbers, loop numbers, areas, and perimeters.

### 4.14. Cell Proliferation Assay

To determine the relative cell proliferation, 1.5  ×  10^4^ cells were plated per well in a 96-well plate and cultured for 24 h in complete cell growth media. The next day cells were exposed to PAH- or HC-EVs (50 μg/mL) for 1 h followed by mock-transfection or transfection of miR-486-5p inhibitor or miR-26a-5p mimic for 48–72 h at 37 °C in a 5% CO_2_-humidified incubator. Post 72 h of transfection, hPAECs proliferation was determined using MTT cell proliferation kit (Sigma-Aldrich, Stockholm, Sweden AB) following manufacturer’s instructions. Data was expressed as percent proliferating cells with respect to the untreated cells.

### 4.15. Western Blot of Analysis

For immunoblot analysis, the following antibodies were used: rabbit monoclonal anti-CD63 (D4I1X; Cell Signaling, Frankfurt am Main, Germany), rabbit monoclonal anti- NF-κB p65 (D14E12; Cell Signaling), rabbit monoclonal anti-PTEN (D4.3; Cell Signaling), rabbit monoclonal anti-TAK1 (D94D7; Cell Signaling), goat polyclonal anti-human VEGF-165 (VEGF-A; Cell Signaling), mouse monoclonal anti–β-actin (Sigma Aldrich, St. Louis, MI, USA), and secondary anti-mouse or anti-rabbit fluorescently labeled antibody (LI-COR). Defined number of hPAECs incubated with EVs and transfected with mock or miR-485-5p inhibitor or miR-26a-5p mimic were lysed in RIPA lysis buffer (Sigma Aldrich) with added proteinase and phosphatase inhibitor cocktail (Cell Signaling). Total protein concentration of the lysate was measured using DC Protein Assay based on the Lowry method (BioRad, Solna, Sweden). Equal amounts of proteins were loaded and separated on 4–20% Tris-Glycine or 4–12% Bis-Tris gradient gels (Invitrogen) by electrophoresis under reducing conditions followed by transfer to PVDF membrane (Merck Millipore, Solna, Sweden). Membranes were blocked and probed overnight at 4 °C with primary antibodies, followed by secondary antibody incubation and washing. Blots were scanned and imaged using the LI-COR system according to the manufacturer’s instructions.

### 4.16. Luciferase Reporter Assay

To quantitatively examine NF-κB activity, luciferase reporter plasmid containing five copies of an NF-κB response element (NF-κB-RE) that drives transcription of a destabilized form of NanoLuc^®^ luciferase (pNL3.2.NF-κB-RE, Promega Biotech AB, Nacka, Sweden) was employed. Cells were first incubated with either HC or PAH EVs for 24 h followed by reversely transfected with miR inhibitor or mimic in a 48-well plate for 24 h. Post miR transfection cells were again transfected with 50 ng pNL3.2.NF-κB-RE luciferase containing plasmid or negative control plasmid without NF-kB-RE for 24 h, before luciferase activity analysis. End-point luminescence was measured using a microplate reader (FLUOstar Omega, BMG LABTECH, Ortenberg, Germany) equipped with Omega Mars software (BMG LABTECH). Data was presented as fold change in luciferase activity as against non-EVs-exposed, control transfected cells.

### 4.17. Statistical Analysis

The two-tailed Student’s *t*-test was used for analysis of two groups, and Two-way analysis of variance (ANOVA) with Newman-Keuls post-test or Tukey’s multiple comparisons test was applied when multiple groups were compared. For non-parametrically distributed data, the two-tailed Mann-Whitney U test was used. Statistical tests were performed using SPSS software, version 23, and GraphPad Prism. A *p*-value of <0.05 was considered statistically significant.

## Figures and Tables

**Figure 1 ijms-23-11964-f001:**
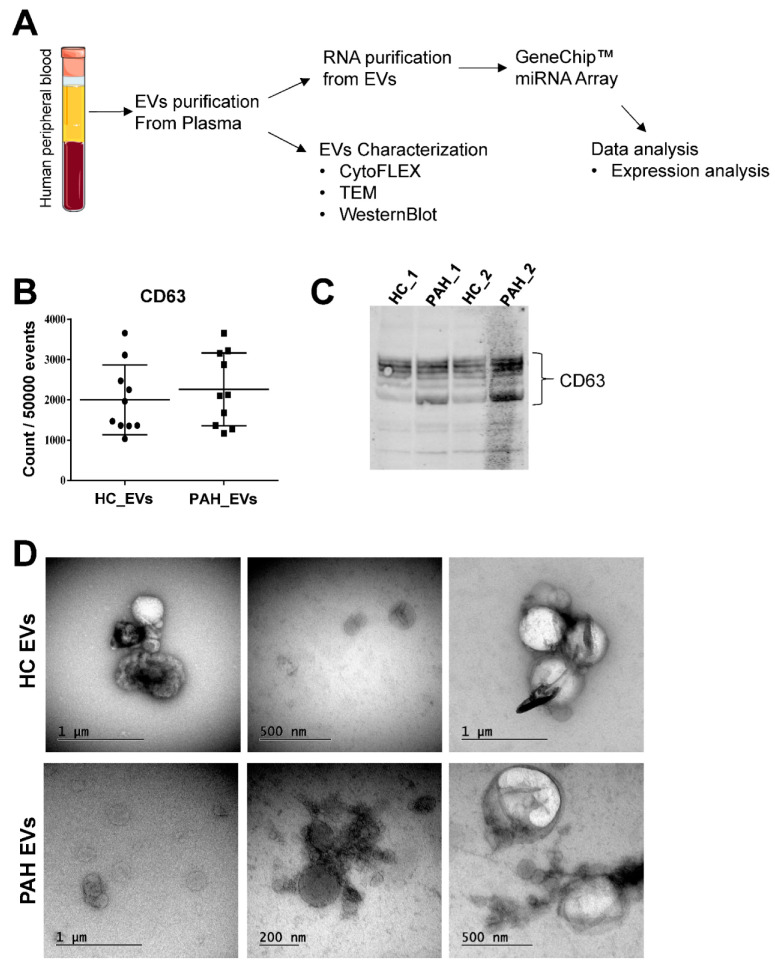
(**A**) Schematic representation of EV preparation and experimental workflow for miRNA profiling and EV characterization. (**B**) Purified EVs from healthy control (HC) and pulmonary arterial hypertension (PAH) patients were subjected to flow cytometry to detect expression levels of exosome-specific marker CD63. Results represent marker count/50,000 events and are given as mean ± S.E.M. (*n* = 10 per group), comparing PAH versus HC. (**C**) Representative immunoblot of CD63 from EVs from HC and PAH patients. (**D**) Representative transmission electron micrographs of purified EVs from HC and PAH patients with uranyl acetate negative staining (scale bar 200–1000 nm) (*n* = 5 per group; 10–20 images were taken per sample).

**Figure 2 ijms-23-11964-f002:**
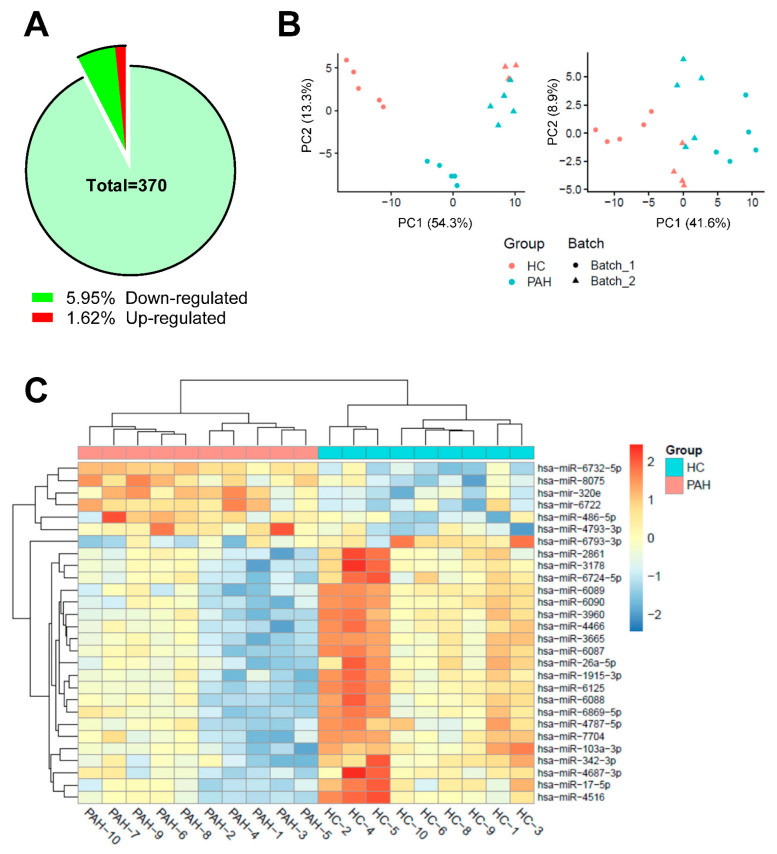
(**A**) Pie diagram depicting the distribution of total miRs and the differentially expressed miRs percentage in PAH EVs as against HC EVs (*n* = 10 for PAH EVs and *n* = 9 for HC EVs). (**B**) Principal component analysis (PCA) before and after batch effect removal using Limma analysis. As depicted in the PCA, the HC (red) and PAH (blue) samples are separated from batch 1 (round) to batch 2 (triangle) (**C**) Heat map of miR microarray expression data from purified EVs of PAH patients (*n* = 10) and HC (*n* = 9). Sample miR species are shown to the right, and individual samples are at the bottom. The heat map depicts significantly different expression levels of miRs and their hierarchical relation among samples. Data is batch corrected, scaled, and centered where red indicates high expression of miR whilst blue indicates relatively lower expression of miR.

**Figure 3 ijms-23-11964-f003:**
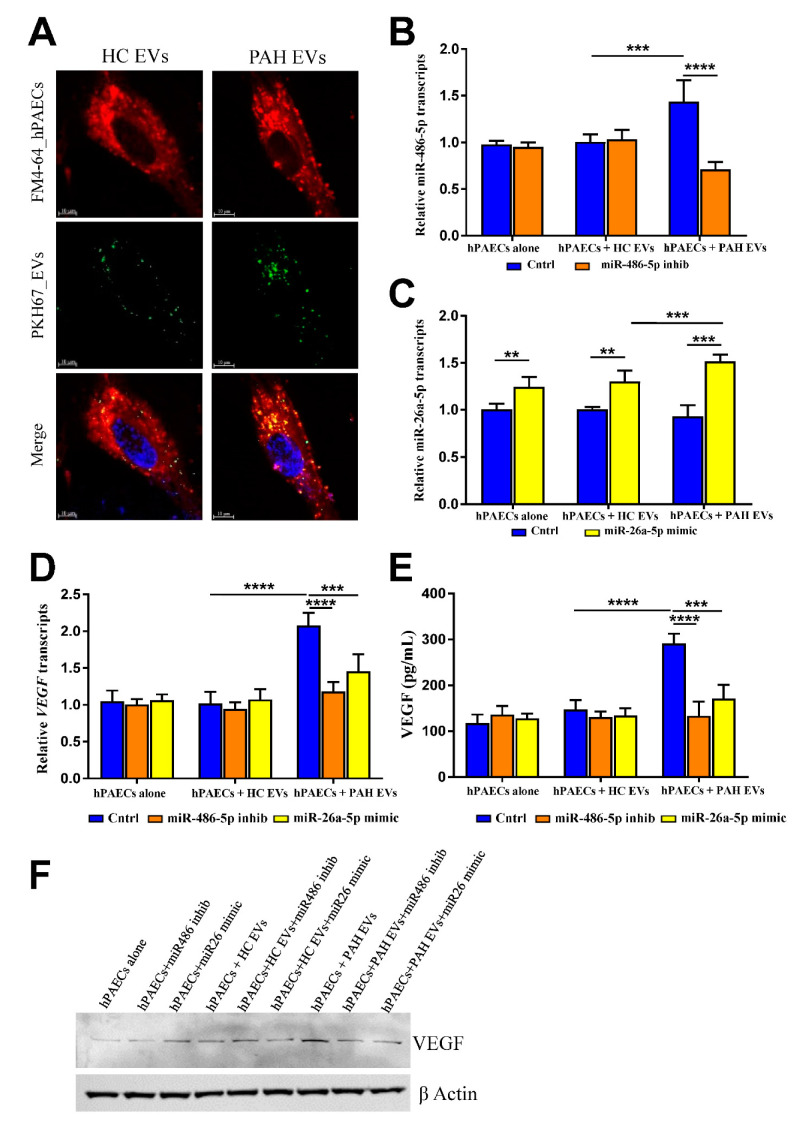
(**A**) HPAECs were exposed to PKH67-labeled EVs for 1 h at 37 °C to facilitate their internalization. After 45 min of incubation cells were labeled with FM4-64FX dye for an additional 15 min followed by fixation, mounting, and confocal analysis. The experiment was performed 3 times, and representative images from at least 15 images per sample are shown. Expression levels of miR-486-5p (**B**) and miR26a-5p (**C**) were quantified using qRT-PCR relative to GAPDH mRNA within hPAECs following miR inhibitor or mimic transfection post EVs incubation. (*n* = 5 per group). (**D**) *VEGF* mRNA transcripts were quantified using qRT-PCR in hPAECs related to its GAPDH housekeeping gene (*n* = 5 per group). (**E**) ELISA analysis to detect the concentrations of proangiogenic VEGF-A in hPAEC culture supernatant pre-incubated with EVs. Results are from *n* = 5 biological replicates and indicated as mean value with SD. (**F**) Representative western blot of VEGF-A together with β-actin loading control on hPAEC lysate pretreated with EVs followed by transfection of miR inhibitor or miR mimic (*n* = 3). Data were analyzed by two-way ANOVA. ** *p* ˂ 0.01, *** *p* < 0.005 and **** *p* < 0.001.

**Figure 4 ijms-23-11964-f004:**
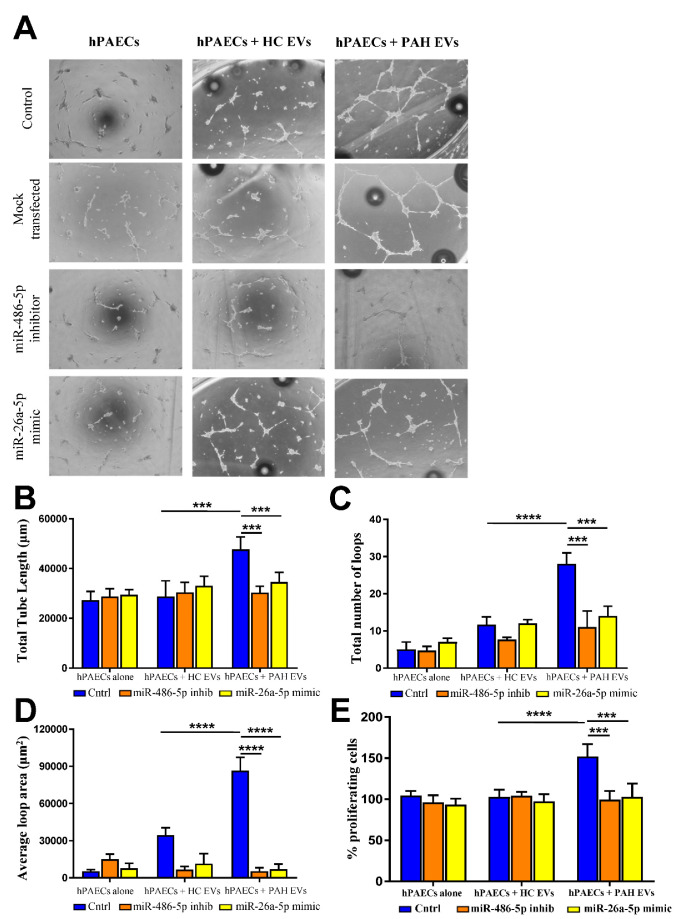
(**A**) Representative photomicrographs of an in vitro Matrigel angiogenesis assay, showing tube-like structure formation by hPAECs, either untreated or incubated with EVs followed by miR inhibitor or mimic transfection. Images were taken at 4× magnification. (**B**) The quantitative analysis of the length of branches. (**C**) Total number of loops. (**D**) Percentage of vessel loop area of in vitro-formed vessel-like structures. Results are from *n* = 3 biological replicates and indicated as mean value with SD. (**E**) Relative cell proliferation of hPAECs presented as % of untreated cells when incubated with PAH or HC EVs for 1 h followed by transfection of miR inhibitor or mimic for an additional 48 h. (*n* = 5 per group in triplicates). Data are presented as mean with SD. All the data were analyzed by two-way ANOVA. *** *p* < 0.005 and **** *p* < 0.001.

**Figure 5 ijms-23-11964-f005:**
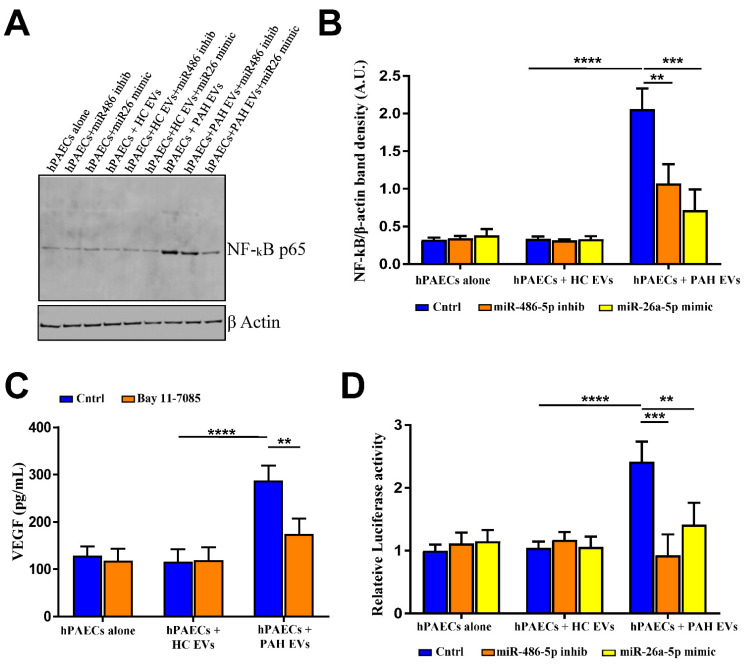
(**A**) Representative immunoblot of NF-kB p65 from hPAECs lysate that was incubated with either HC or PAH EVs followed by miR inhibitor or mimic transfection. (*n* = 3). (**B**) Densitometric quantitation of NF-kB band intensity from blots from 5A. Data are presented as ratios of band densities of NF-kB to b-actin in arbitrary units (A.U.) from three independent experiments and given as mean with SD. (**C**) ELISA analysis to detect the concentrations of proangiogenic VEGF-A in hPAEC culture supernatant pre-incubated with EVs in the presence or absence of NF-kB inhibitor (Bay 11-7085; 50 nM for 24 h). Results are from *n* = 3 biological replicates and indicated as mean value with SD. (**D**) Luciferase reporter quantification of NFkB activation in hPAECs was done in double transfected cells that were pre-incubated or not with EVs prior to mock, miR inhibitor, or mimic transfection followed by reporter transfection. Luciferase luminescence is represented as a fold change of luciferase activity compared to mock-transfected cells and is given as mean with SD (*n* = 5 per group). *p*-value by two-way ANOVA. ** *p* ˂ 0.01, *** *p* < 0.005, **** *p* < 0.001.

**Table 1 ijms-23-11964-t001:** List of significantly altered miRs obtained from microarray analysis of PAH EVs (*n* = 10 when compared to their HC counterparts (*n* = 9).

Transcript	Type	Log Fold Change	Adj. P-Value
**Down-regulated**			
hsa-miR-6089	miRNA	−2.78175	0.003676
hsa-miR-3665	miRNA	−2.5509	0.006932
hsa-miR-6090	miRNA	−2.54716	0.003718
hsa-miR-6087	miRNA	−2.13755	0.01727
hsa-miR-6125	miRNA	−2.1033	0.005246
hsa-miR-26a-5p	miRNA	−2.07892	0.009452
hsa-miR-7704	miRNA	−1.9402	0.029668
hsa-miR-4466	miRNA	−1.86555	0.009452
hsa-miR-6088	miRNA	−1.85115	0.01727
hsa-miR-4516	miRNA	−1.73597	0.031291
hsa-miR-103a-3p	miRNA	−1.73382	0.006913
hsa-miR-6869-5p	miRNA	−1.68796	0.031291
hsa-miR-1915-3p	miRNA	−1.62478	0.011043
hsa-miR-3960	miRNA	−1.55252	0.00461
hsa-miR-342-3p	miRNA	−1.1534	0.009452
hsa-miR-17-5p	miRNA	−1.12446	0.017975
hsa-miR-2861	miRNA	−1.07266	0.049252
hsa-miR-3178	miRNA	−0.85985	0.029668
hsa-miR-4687-3p	miRNA	−0.82483	0.049252
hsa-miR-6724-5p	miRNA	−0.7889	0.033406
hsa-miR-4787-5p	miRNA	−0.74063	0.031291
hsa-miR-6793-3p	miRNA	−0.40702	0.048881
**Up-regulated**			
hsa-mir-320e	Stem-loop	0.379859	0.009452
hsa-mir-6722	Stem-loop	0.380027	0.018802
hsa-miR-8075	miRNA	0.402266	0.01727
hsa-miR-6732-5p	miRNA	0.561575	0.000734
hsa-miR-4793-3p	miRNA	0.730484	0.029668
hsa-miR-486-5p	miRNA	1.82015	0.03513

## Data Availability

Not applicable.

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
