# Peer review of "MicroRNA in Extracellular Vesicles from Patients with Pulmonary Arterial Hypertension Alters Endothelial Angiogenic Response"

_ijms, 2022, doi:10.3390/ijms231911964_

Round 1

Reviewer 1 Report

The authors have investigted micro RNA in circulating vesicles of patients with pulmonary arterial hypertension and found two altered miRNA expression, miR-486-5p and miR-26a-5p.

The manuscript is of interest for possible understanding of disease development or progression and better treatment of patients with pulmonary arterial hypertension. Some questions remain . 

1) In the PAH EV experiments, were the effects of miR-486-5p inhibition and overexpression of miR-26a-5p seen as an effect caused by altered miRNA expression of these two candidates as a singular appoach, or did modulation of both miR candidates at the same time show additive or synergistic effects ?  

2) Several patients with severe PAH requiring intensive care and mechanical ventilation receive nitric oxide treatment. Will the pathways shown involving the two miRNAs also modulate the nitric oxide effects or pathway ? 

3) Will the effects of the miRNAs shown in PAH EVs indicate immediate and early effects or midterm alterations caused by inhibition of cellular proangiogenic and proliferative responses ? 

4) In the abstract the authors mention that the miR profile can be targeted in order to restrict angiogenesis and reduce pulmonary entotheliun activation. Please give an idea how this could be achieved in an animal experiment setting or in the clinical situation.

5) Do the authors plan clinical investigations to look for miRNA expression in patients with PAH regarding the altered expression of the two miRNAs and look an correlation to disease pogression and outcome ?  

Author Response

We thank the reviewer for her/his time to critical evaluate our work. Reviewer's insights have improved the manuscript quality and our own understanding in this regard. Please find attached our detailed response to your queries.

Best regards.

Reviewer 2 Report

The manuscript "MicroRNA in extracellular vesicles from patients with pulmonary arterial hypertension alters endothelial angiogenic response" by Khandagale et al. is an interesting study on a novel mechanism of endothelial dysfuction in pulmonary hypertension.  The manuscript will add valuable information to the pulmonary hypertension field. However, the paper is missing a few key experiments. 1: VEGF transcripts are elevated by PAH EVs and reduced by the miRNAs, but western blot data of actual protein levels is missing. 2: The author's previous paper showed a dramatic effect of PAH EVs on FGF levels. Do the miRNAs in this study prevent FGF expression increases? 3: The data suggest that NF-kappaB signaling mediates the pro-angiogenic effect of PAH EVs. This idea would be strengthened by performing matrigel angiogenesis assays or VEGF expression assays +/- PAH EVs with or without NF-kappaB inhibitors or siRNA knockdown. These additions would strengthen the paper, further supporting the main findings.

Author Response

We deeply appreciate reviewer's time in evaluating and critically deliberating our work. This has improved the quality and help adding further important data to this manuscript.

Please see attached our point wise detailed response to your queries.

Best regards.

Round 2

Reviewer 2 Report

The authors have now included several key experiments that strengthen the proposed mechanism, which involves miRNA modulation of NF-kappaB signaling to increase VEGF transcription and protein expression. The paper is a very interesting contribution to the field of vascular effects of EVs and the pathophysiology of PAH.